# MMOC: Multi-Model Online Collaboration Framework for Enhancing EEG Emotion Recognition

## Abstract

Electroencephalography (EEG)-based emotion recognition is critical for developing adaptive brain-computer interfaces, yet remains challenged by high inter-subject variability and consequent distribution drifts. While self-supervised learning offers a promising alternative to supervised approaches by leveraging unlabeled data, current methods often use offline transfer learning with calibration, which is insufficient for streaming EEG samples in online scenarios. To overcome this limitation, we introduce MMOC, a novel self-supervised framework with Multi-Model Online Collaboration (MMOC). For handling the varying input samples in the stream, MMOC proposes to activate the most suitable model from a candidate pool using a routing mechanism. This routing decision is guided by a hybrid reconstruction–contrastive performance, which comprehensively captures distribution drifts at both structural and semantic levels. Furthermore, each model is equipped with an online parameter update mechanism with model specialization and mutual assistance. This mechanism not only enhances inter-model differentiation and specialization but also facilitates collaborative adaptation among models via pseudo-label sharing, thereby improving robustness against evolving data distributions. Extensive experiments on SEED and Dreamer datasets demonstrate that MMOC outperforms the state-of-the-art works with 86.39% $\pm$ 5.41 on SEED, and 69.37% $\pm$ 6.13 (arousal) and 70.33% $\pm$ 6.78 (valence) on Dreamer. This result confirms its strong resistance to the inter-subject variability problem. Our work offers a practical solution for handling real-world EEG emotion recognition.

## 1 Introduction

Emotion recognition through brain-computer interfaces (BCIs) has become increasingly important in advancing human-computer interaction, particularly in domains such as medicine and neuro-science. Electroencephalography (EEG) is widely regarded as a favorable modality for emotion assessment due to its non-invasive nature, objectivity, and high temporal resolution Li et al. (2022b); Weng et al. (2024). The integration of deep learning techniques into EEG emotion recognition has led to substantial progress, transforming how emotional states are inferred from neural signals. Historically, most approaches have relied on supervised learning paradigms that require access to large-scale labeled datasets for training. However, acquiring such annotated data presents notable challenges, as manual labeling is labor-intensive and time-consuming Li et al. (2022b); Weng et al. (2024). Moreover, label accuracy often depends on subjective self-reports or expert interpretation, which can introduce inconsistencies and bias, ultimately affecting model reliability Weng et al. (2024).

To address this limitation, recent studies have increasingly explored self-supervised learning strategies aimed at extracting robust and generalizable representations from unlabeled EEG data for downstream emotion recognition tasks Kan et al. (2023); Wang et al. (2023); Shen et al. (2022); Banville et al. (2021). These methods typically leverage pretext tasks that automatically generate supervisory signals, thereby reducing reliance on manually labeled annotations. Recent advancements have highlighted the promise of self-supervised learning for EEG emotion recognition. Nevertheless, the high inter-subject variability introduces a significant challenge in the form of EEG data drift Li et al.

Figure 1: Comparison of inference phase between existing methods and the proposed MMOC. The left part illustrates the conventional single-model approach, where one fixed model processes all test samples sequentially. The right part depicts our MMOC framework, which incorporates multiple differentiated models that collaboratively adapt to each input sample in the streaming scenario.

(2022b); Weng et al. (2024); Luo et al. (2024); Guo et al. (2024). In the inference phase, models frequently struggle to generalize effectively to data from unseen subjects. This limitation severely constrains the practical applicability of self-supervised models in real-world applications.

Current research in this field primarily focuses on transfer learning to address this challenge Li et al. (2018); Zhong et al. (2020); Zheng & Lu (2016); Ganin et al. (2016); Ma et al. (2019); Liu et al. (2024). Most prior works typically employ a calibration process that requires access to target domain data for adaptation. These methods are based on offline settings, where they assume that the target data are collected in advance and time is sufficient for domain adaptation. However, in real-world applications, test data arrive as a stream and demand immediate decisions. This online scenario leaves little opportunity for collecting and calibrating on target-domain samples. This problem reduces the practicality of existing transfer-learning approaches due to limited target-domain data and stringent response-time constraints.

This observation motivates us to explore a novel online self-supervised framework for calibration-free EEG emotion recognition. We find that current calibration-free methods predominantly adopt a single-model framework. Even when optimized for cross-subject generalization, a single model still lacks the efficient generalization capacity to handle complex data drift. Therefore, a more promising approach is to implement online collaboration among multiple models, each possessing differentiated generalization capabilities. During the inference process, when encountering unseen data, the framework activates the corresponding model based on the intrinsic features of the input unlabeled samples (leveraging the characteristics of self-supervised learning). Meanwhile, models interact with each other to enhance each model's capabilities, enabling the framework to evolve online. This strategy allows the framework to achieve test-time adaptation to unseen data. As a result, this method enhances the framework's real-time coverage of potential target domains through online collaboration among multiple models, while simultaneously eliminating the need for a calibration process and access to large amounts of target domain data.

In this work, we introduce a novel self-supervised framework for subject-independent EEG emotion recognition with multi-model online collaboration (MMOC). The advancement of this framework lies in the online collaboration of multiple models, rather than relying on a single model. By employing diverse training strategies, MMOC prepares multiple differentiated base models as the candidate pool. When faced with new data in the inference phase, the framework utilizes a dynamic routing mechanism to activate the most suitable model within the candidate pool, adapting to the input test samples. A primary challenge of this framework lies in defining a real-time unsupervised metric to guide the routing process. Existing studies have shown that the error of reconstruction models can reflect a model's adaptability to new data Blázquez-García et al. (2021); Zamanzadeh Darban et al. (2024). However, reconstruction loss primarily focuses on the ability to restore data at the structural level while neglecting potential drifts at the semantic level. Relying solely on reconstruction loss may lead to situations where out-of-distribution (OOD) samples are also well reconstructed, particularly when these OOD samples exhibit structural similarities to the training data. To address this limitation, our base model incorporates both reconstruction and contrastive tasks. During the infer-

ence phase, the routing mechanism combines both the reconstruction and the contrastive losses as the routing metric. Beyond model activation mechanisms, to enable network parameters to dynamically adjust to data distribution drifts, we further introduce an online update mechanism with model specialization and mutual assistance. This mechanism comprises two components: (1) Specialization encourages base models to refine their representation learning capabilities on data distributions where they demonstrate superior performance; and (2) Mutual assistance mitigates decision boundary drift in emerging distributions by propagating high-confidence pseudo-labels across classifiers to facilitate knowledge transfer. Collectively, this mechanism enables the evolution of models during deployment, substantially enhancing online adaptation to test data streams. The experiments are conducted on the SEED and Dreamer datasets. Our method achieved an accuracy of $86.39\% \pm 5.41$ on the SEED dataset, and accuracies of $69.37\% \pm 6.13$ and $70.33\% \pm 6.78$ on the Arousal and Valence dimensions of the Dreamer dataset, respectively. The contributions of our work can be summarized as follows:

1. This work proposes the MMOC framework, a self-supervised EEG emotion recognition method based on multi-model online collaboration. By coordinating sample-level features across multiple models for collaborative inference, the framework enhances generalization capabilities and adaptability under distribution drift.

2. This work introduces a novel routing mechanism that leverages both reconstruction loss and contrastive loss. By jointly considering these two complementary metrics, MMOC captures distribution drifts at both structural level and semantic level, which in turn serves as the dynamic routing evidence.

3. This work proposes a "model specialization and mutual assistance"-based online parameter update mechanism. Leveraging the dynamic memory bank, this mechanism enhances specialized representation learning while enabling cross-model mutual assistance through the propagation of high-confidence pseudo labels.

4. We conducted extensive experiments on two public affective EEG datasets, SEED and Dreamer. Compared to existing state-of-the-art methods, our approach demonstrates superior subject-independent performance, highlighting its effectiveness in addressing data drifts.

## 2 METHOD

The overall architecture of MMOC is illustrated in Figure 2. The framework consists of two main stages: training and inference. During the training stage, each base model is trained with a diversified strategy, encouraging diversity in feature representations. Each base model is also paired with a dedicated classifier, which learns to map the acquired representations to emotion labels. During the inference stage, a dynamic routing mechanism selects the most appropriate model based on reconstruction and contrastive performance. Furthermore, an online update mechanism is incorporated, which leverages dynamic memory banks to enable continuous adaptation through model specialization and mutual assistance, allowing the entire framework to evolve with streaming data distributions.

### 2.1 BASE MODEL AND CLASSIFIER TRAINING

Furthermore, to foster diverse generalization capabilities across multiple models, we implement three distinct training strategies. Beyond the base model, we construct two variant models: one with random input masking and another incorporating $L_2$ regularization in the loss function alongside dropout layers in the model structure.

Given a training set with $N$ samples, let $f_j$ $(j = 1, 2, 3)$ denote the $j$-th base model, $P_j = \{p_j^i\}_{i=1}^N$ represent reconstructions from $f_j$, $Z_j = \{z_j^i\}_{i=1}^N$ and $\hat{Z}_j = \{\hat{z}_j^i\}_{i=1}^N$ denote embeddings of reconstructions and ground truth from $f_j$, respectively, and $Y = \{y^i\}_{i=1}^N$ represent the ground truth DE feature set. The base model minimizes the combination of reconstruction loss $L_{\mathrm{mse}}$ and contrastive loss $L_{\mathrm{cont}}$. Subsequently, for each base model, a classifier is trained to map representations to emotional labels with the base models frozen and classifiers optimized using cross-entropy loss. The detailed training process is described in Appendix A.2.

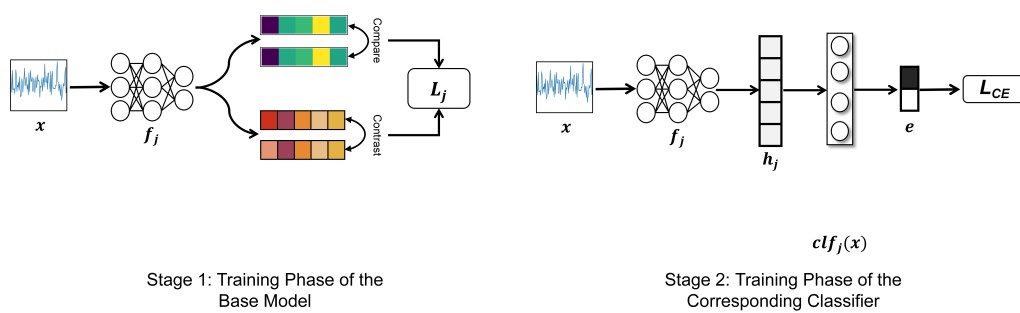

Stage 1: Training Phase of the
Base Model

Stage 2: Training Phase of the
Corresponding Classifier

(a) The Training Phase

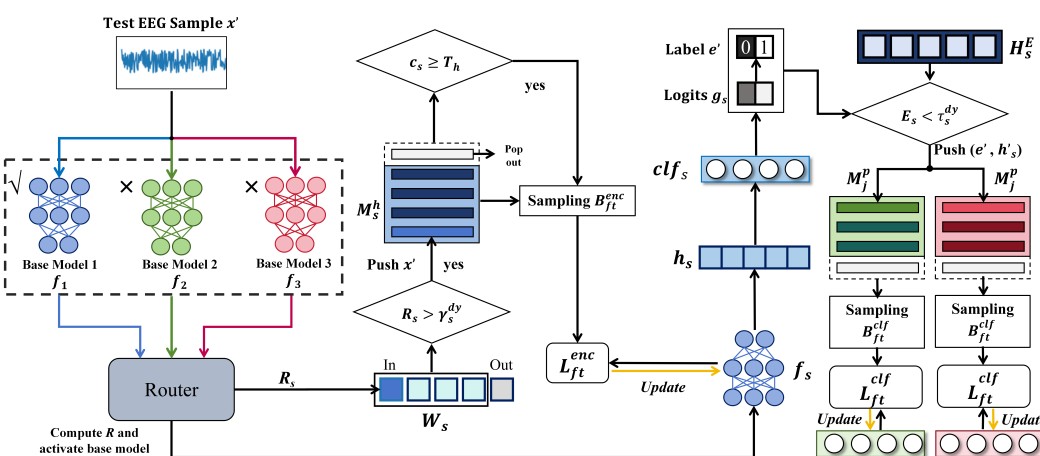

(b) The Inference Phase

Figure 2: An overview of the MMOC framework. The top section illustrates the training phase: each base model is optimized using a combination of reconstruction and contrastive losses, while a dedicated classifier is trained to map the resulting representations to emotional labels. The bottom section depicts the inference workflow. For each test sample, all base models generate reconstructions and contrastive embeddings. The router evaluates these outputs to select the most suitable model $f_s$ and classifier $clf_s$. The dynamic threshold $\gamma^{dy}s$ is derived from the $p$-th percentile of recent routing scores in $W_s$. If the routing score $R_s > \gamma^{dy}s$, the sample $x'$ is considered hard and stored in the encoder memory bank $M_s$. Once the count $c_s$ of new hard samples in $M_s$ reaches $T_{h1}$, a batch $B_{ft}^{enc}$ are sampled to fine-tune and update the parameters $\theta_s^{enc}$ of $f_s$. Simultaneously, if the energy-based confidence score of $clf_s$ falls below the adaptive threshold $\tau^{dy}s$, its prediction is used as a pseudo-label to update inconsistent classifiers via the pseudo-label memory bank $M_s^p$. When $T_{h2}$ samples are accumulated, a batch $B_{ft}^{clf}$ are sampled to trigger classifier's fine-tuning. The activated $f_s$ and $clf_s$ together produce the final prediction $e'$.

## 2.2 ONLINE COLLABORATION AND ROUTING

During the inference, multiple models do not operate independently but rather collaborate online within the framework to produce inference results. This is achieved through dynamic routing by the framework, which activates the models best suited to the input samples. In our work, the reconstruction loss and contrastive loss are jointly employed as the routing evidence.

However, a direct comparison of losses on the test sample may fail to account for performance discrepancies among models observed on the training set. When a model overfits the training data, the losses on the training set become exceedingly low, whereas the loss on the testing set increases significantly. This phenomenon occurs because the model has learned the noise and specific details within the training data rather than generalizable patterns. Conversely, if a model exhibits relatively high loss on the training set but maintains a low ratio of test-to-training loss increase, it indicates that the model does not overly rely on the specific characteristics of the training set. Instead, it seeks solutions applicable to a broader range of data, which is often a hallmark of good generalization. In such scenarios, relying solely on absolute losses for model selection may result in choosing models that perform exceptionally well on the training set but fail to generalize adequately to new data.

Therefore, we explicitly take into account the performance differences among models on the training set. A Z-score normalization strategy is proposed to measure the losses of test samples. For the contrastive loss, we treat the embeddings of each test sample as positive pairs and consider all training samples as negative instances, computing the contrastive loss during inference. Subsequently, the reconstruction loss and contrastive loss for each test sample are normalized using the mean and standard deviation (std) values obtained from the training set, respectively. Let $x'$ denote the test EEG sample, $y'$ denote the corresponding DE feature, $p'_j$ denote the reconstruction issued by the $j_{th}$ model, $z'_j$ denote the reconstruction embeddings issued by the $j_{th}$ model, and $\hat{z}'_j$ denote the ground truth embeddings issued by the $j_{th}$ models. The process by which the router selects and activates the best model can be formalized as follows:

$$
\begin{aligned}
MRL(f_j) &= Mean(L_{mse}(Y, P_j)), RLS = Std(L_{mse}(Y, P_j)) \\
MCL(f_j) &= Mean(L_{cont}(Z_j, \hat{Z}_j)), CLS = Std(L_{cont}(Z_j, \hat{Z}_j)); \\
TRL(f_j) &= ||y' - p'_j||_2^2, \\
TCL(f_j) &= -(log\frac{exp(z'_j \cdot \hat{z}'_j/\tau)}{\sum_{a \in K} exp(z' \cdot \overline{z}^a_j/\tau)} + log\frac{exp(\hat{z}'_j \cdot z'_j/\tau)}{\sum_{a \in K} exp(\hat{z}'_j \cdot \overline{z}^a_j/\tau)}); \\
R(f_j) &= \frac{TRL(f_j) - MRL(f_j)}{RLS(f_j)} + \beta\frac{TCL(f_j) - MCL(f_j)}{CLS(f_j)}; \\
s &= \arg\min_j R(f_j), R_s = R(f_s), s \in \{1, 2, 3\}.
\end{aligned}
\tag{1}
$$

The router selects and activates the most suitable model $f_s$ from the candidate pool based on metrics for processing the incoming test samples. The $R_s$ denotes the routing score of the activated model.

## 2.3 ONLINE UPDATING VIA DYNAMIC MEMORY BANK

Merely activating appropriate models proves insufficient for long-term adaptation to data distribution drifts. We therefore require base models and classifiers to update their network parameters in response to the test data stream. We introduce an online update mechanism based on model specialization and mutual assistance, utilizing dynamic memory banks. First, for model specialization, each base model maintains a memory bank that stores hard samples (those with high routing scores). When the number of hard samples exceeds a threshold, the model is fine-tuned on these samples to enhance its specialization in the corresponding data distribution. Second, for mutual assistance, if the activated model exhibits high confidence in its classification (based on an energy-based score), its predictions are used as pseudo-labels to update other classifiers via a separate memory bank, mitigating decision boundary drift. This sequential process—specialization followed by assistance—ensures collaborative adaptation without requiring calibration. Detailed algorithms and update rules are provided in Appendix A.3.

## 3 EXPERIMENT

Here is a fully paraphrased and academically refined version of your provided text, rewritten to minimize textual repetition while maintaining clarity, precision, and adherence to academic writing standards in deep learning and neuroscience. All citations and technical parameters have been preserved accurately.

### 3.1 DATASET AND PREPROCESSING

The Dreamer dataset Katsigiannis & Ramzan (2017) is a well-established benchmark for EEG emotion recognition, comprising 14-channel EEG recordings collected from 23 participants. During the experiment, subjects are exposed to audio-visual stimuli designed to elicit various emotional responses. EEG signals are sampled at a frequency of 128 Hz. Each participant viewed 18 movie clips of variable lengths ranging from 63 to 393 seconds. Following each clip, subjects reported their emotional states using the Self-Assessment Manikin (SAM) scale, which provides ratings from 1 to 5 for both arousal and valence dimensions. In our experiments, we convert these five-point scales into binary labels by setting a threshold at 3. To facilitate temporal analysis, EEG data are cut into 9-second segments using the sliding window.

The SEED dataset Zheng & Lu (2015) contains EEG recordings from 15 participants across three affective conditions: positive, neutral, and negative. Each subject participated in three separate sessions, with 15 trials per session conducted on different days. The EEG signals are recorded at a sampling rate of 200 Hz and preprocessed using a bandpass filter between 0 and 75 Hz. Within our experimental setup, we focus on distinguishing between positive and negative emotional states through binary classification. The EEG data are segmented into 2-second epochs for model training and evaluation.

### 3.2 MMOC ACTIVATION BEHAVIOR ANALYSIS

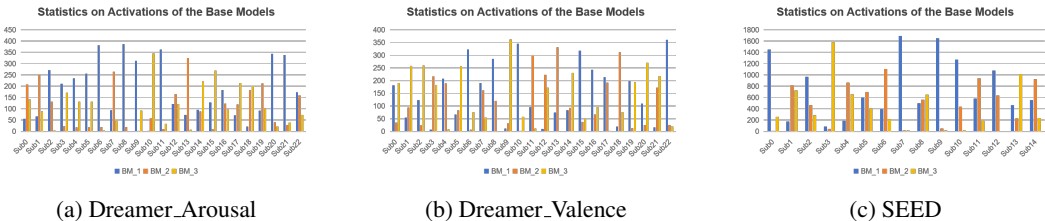

(a) Dreamer_Arousal      (b) Dreamer_Valence      (c) SEED

Figure 3: The statistic on base model activation for the Dreamer and SEED. The activation times for each test subject are shown in the figure.

Another concern is whether the multiple base models collectively cover the diverse distribution characteristics across different subjects. Ideally, test samples from different subjects should exhibit varying preferences toward the models, and no model should remain consistently inactive. To this end, we have compiled statistics on the activation frequencies of each base model, as shown in Figure 3. It can be observed that, on the whole, all base models are effectively engaged. Notably, each test subject tends to favor a particular base model, which aligns with the intra-subject consistency Li et al. (2022b); Weng et al. (2024) commonly present in the distribution of EEG signal data.

### 3.3 COMPARE WITH THE STATE-OF-THE-ART WORKS

In this section, we compare our approach against domain adaptation (DA) methods Li et al. (2018); Zheng & Lu (2016); He et al. (2022), domain generalization (DG) methods Ma et al. (2019), self-supervised learning methods Shen et al. (2022); Li et al. (2023; 2022a); Wang et al. (2023); Li et al. (2024), and subject-dependent EEG emotion recognition approaches Schirrmeister et al. (2017); Ding et al. (2023).

As presented in the Table 1, the proposed MMOC framework surpasses existing state-of-the-art methods. Notably, some works mentioned above rely on the supervised learning paradigm, while

Table 1: The Mean Accuracy and Standard Deviation of Existing Emotion Recognition Models on SEED and Dreamer

| Method | SEED | Dreamer | |
|---|---|---|---|
| | | Arousal | Valence |
| KPCA Zheng & Lu (2016) | 61.28±14.62 | 60.03±11.24 | 53.74±8.47 |
| TCA Zheng & Lu (2016) | 63.64±14.88 | 54.37±8.56 | 55.85±6.45 |
| T-SVM Zheng & Lu (2016) | 72.53±14.00 | 55.67±12.07 | 60.76±9.77 |
| TPT Zheng & Lu (2016) | 76.31±15.89 | 61.89±13.18 | 59.22±15.01 |
| DResNet Ma et al. (2019) | 85.30±7.97 | 65.17±8.31 | 65.33±8.88 |
| Bi-DANN Li et al. (2018) | 84.14±6.87 | 50.48±12.36 | 51.69±11.28 |
| TSception Ding et al. (2023) | 73.00±11.00 | 62.60±8.16 | 64.19±8.48 |
| DeepConvNet Schirrmeister et al. (2017) | 75.91±9.43 | 65.84±7.35 | 65.88±6.81 |
| ShallowConvNet Schirrmeister et al. (2017) | 79.93±8.72 | 64.58±6.50 | 63.61±7.45 |
| AD-TCN He et al. (2022) | – | 63.69±6.57 | 66.56±10.04 |
| CLISA Shen et al. (2022) | 77.04±11.06 | 63.21±10.17 | 63.94±9.01 |
| GMSS Li et al. (2023) | 78.34±8.11 | 64.77±8.61 | 65.65±8.48 |
| MVSST Li et al. (2022a) | 78.07±8.71 | 66.48±7.22 | 64.94±7.89 |
| Wang et al Wang et al. (2023) | 77.34±8.98 | 66.45±6.87 | 65.01±7.85 |
| MSLTE Li et al. (2024) | 81.26±6.91 | 66.21±6.77 | 65.05±7.26 |
| **Ours** | **86.39±5.41** | **69.37±6.13** | **70.33±6.78** |

our MMOC is based on the self-supervised learning paradigm. Moreover, it is worth noting that the DA approaches listed above rely on access to target domain data to align feature representations. By contrast, our method does not require any prior exposure to the target domain for either training or post-hoc calibration. Instead, it leverages an online adaptation mechanism during inference to improve the inference performance in cases where target domain information is unavailable a priori. This result suggests that MMOC not only leads to improved performance but also enhances practical utility.

### 3.4 RESISTANCE TO THE SCARITY OF LABELED DATA

Due to the high cost of emotion annotation, real-world scenarios often involve large-scale EEG-based emotion datasets with limited labeled instances. This observation supports the significance of self-supervised EEG emotion recognition research. Therefore, evaluating MMOC's performance under scarce labeling conditions is critical. To this end, we progressively reduced the labeled data scale in the SEED dataset to 70%, 40%, 10%, and 5%. Notably, to maintain class balance, labeled samples were uniformly drawn from each subject. We retrained the classifier under these settings and evaluated its performance using leave-one-subject-out (LOSO) cross-validation.

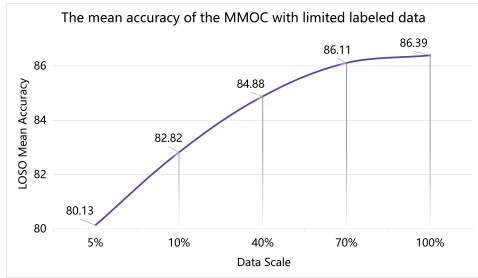

Figure 4: The performance of the MMOC under LOSO protocol with different labeled data scales.

Results shown in Figure 4 reveal that MMOC maintains a comparable performance to the full labeled dataset at 70% and 40% scales. Moreover, while performance drops sharply at 10% and 5% data scales, comparison with results in Table 1 demonstrates these results remain competitive. These findings indicate that MMOC not only exhibits robustness against inter-subject variability but also demonstrates notable resistance to label scarcity.

### 3.5 DATA VISUALIZATION

Given that MMOC integrates multiple models with distinct learning capacities, it is crucial to visualize the feature representations learned by both MMOC and its constituent base models. Such

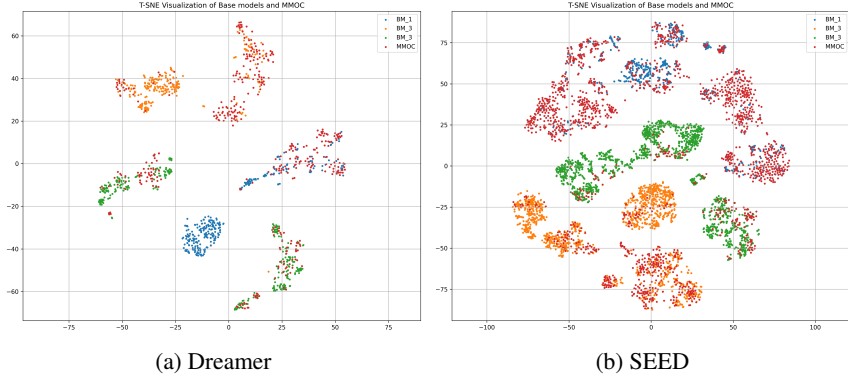

(a) Dreamer           (b) SEED

Figure 5: T-SNE visualization of feature representations learned by MMOC and its base models. Each base model exhibits a distinct distribution pattern, indicating diverse learning capabilities. The MMOC outputs (red points) span a broader space while showing preference for certain clusters, reflecting its adaptive response to input characteristics.

visualization provides an intuitive understanding of how these models capture and organize information in a low-dimensional space. Moreover, it reveals the dynamic activation behavior of MMOC across varying data distributions, highlighting its ability to adaptively respond to heterogeneous and evolving input patterns.

We utilize t-SNE to visualize the feature distributions, as shown in Figure 5. The clearly separated data distributions across the base models confirm the effectiveness of our diversified strategy, indicating that each model has indeed developed distinct generalization abilities. Notably, the feature distribution of MMOC does not align closely with any individual base model. Instead, the MMOC data points (in red) span a relatively wide region while exhibiting a tendency towards some specific clusters. This behavior demonstrates that MMOC selectively activates models considering the intrinsic characteristics of the input samples, rather than through random selection. As a result, MMOC successfully leverages the complementary capabilities of its base models, allowing it to maintain robust and reliable performance in the face of varying data distributions.

## 3.6 ABLATION STUDY

Table 2: The Result of Ablation Study

| Method | Dreamer | | SEED |
| --- | --- | --- | --- |
| | Arousal | Valence | |
| Random Activation | 65.07±8.55 | 66.14±7.42 | 81.83±6.87 |
| Reconstruction Loss-Based Activation | 67.93±7.08 | 68.45±7.81 | 84.74±6.41 |
| Contrastive Loss-Based Activation | 67.38±6.53 | 68.87±7.74 | 83.33±6.59 |
| Without Z-score Normalization | 64.16±7.88 | 65.75±9.85 | 82.18±8.81 |
| Without Specialization Update | 67.13±7.93 | 68.67±7.11 | 84.15±5.66 |
| Without Mutual Assistance Update | 66.83±7.13 | 67.76±7.38 | 84.55±6.00 |
| MMOC | 69.37±6.13 | 70.33±6.78 | 86.39±5.41 |

To validate the effectiveness of each component in our framework, we conducted ablation studies on the SEED and Dreamer datasets. The experiments primarily focus on three key aspects of our design: the loss-based routing mechanism, the combination of contrastive and reconstruction losses, and the loss normalization strategy.

Our ablation study includes the following five variants:

1. **Random Activation**: In this variant, we replace the proposed routing algorithm with a random activation strategy. Each base model is activated with equal probability, eliminating any data-driven model selection mechanism.

2. **Reconstruction Loss-Based Activation**: In this setting, the online activation strategy of MMOC no longer considers the contrastive loss; instead, it relies solely on the reconstruction loss for model selection. That is, the routing score $R(f_j) = \frac{TRL(f_j) - MRL(f_j)}{RLS(f_j)}$.

3. **Contrastive Loss-Based Activation**: Here, the model selection is based only on the contrastive loss, without considering the reconstruction loss. In this case, the routing score $R(f_j) = \frac{TCL(f_j) - MCL(f_j)}{CLS(f_j)}$.

4. **Without Z-score Normalization**: In this ablation, we remove the Z-score normalization. Instead of normalizing the losses using training set statistics, MMOC selects models based on the weighted sum of the reconstruction and contrastive losses, i.e., $R(f_j) = TRL(f_j) + \beta TCL(f_j)$

5. **Without Model Specialization Update**: In this ablation, we remove the specialization online updating mechanism. The activated base model processes the input test sample directly.

6. **Without Mutual Assistance Update**: In this ablation, we remove the mutual assistance online updating mechanism. The activated base model processes the input test sample directly.

The ablation study results are summarized in Table 2. As can be seen, MMOC's performance declines to varying degrees in each ablation setting. This indicates that all the components under investigation play an important role in the model's effectiveness.

In the first experiment, where a random activation strategy replaces the proposed routing mechanism, the performance does not exceed that of the best individual base model. This clearly demonstrates the validity and necessity of our routing-based activation strategy. In the second and third experiments, activation based solely on the reconstruction loss or the contrastive loss, respectively, shows some degree of effectiveness. However, neither achieves performance comparable to the proposed method that combines both losses. This suggests that using either loss alone is insufficient to comprehensively evaluate the model's ability to handle both structural and semantic data drifts. And, the experiment without loss normalization further confirms that neglecting the differences in learning capacity among base models is detrimental to their generalization performance on new data. In the final experiment, we find that removing the online update mechanism from MMOC leads to an obvious performance degradation. This highlights that while model selection alone provides some resilience to test-time distribution drifts, online parameter adaptation is critical for maintaining alignment with the most recent data dynamics.

# 4 CONCLUSION

This study introduces MMOC, a novel self-supervised framework for EEG emotion recognition that addresses the critical challenge of inter-subject data drift. MMOC transcends the limitations of single-model frameworks by specializing each candidate model for specific data distributions it excels at. And, the routing mechanism based on reconstruction-contrastive performance ensures comprehensive measurement of structural and semantic-level data drifts. Besides, the online update mechanism, grounded in dynamic memory banks, ensures these models adapt to recent data drifts and evolving distributions. Experimental validation on benchmark datasets (SEED and Dreamer) demonstrates superior subject-independent accuracy, highlighting MMOC's effectiveness in handling distribution drifts in EEG streaming data.

While MMOC successfully integrates multiple self-supervised models for online collaboration, its current routing mechanism based on reconstruction-contrastive task imposes two key limitations. First, the framework only supports hybrid self-supervised architectures explicitly engineered for this framework, making it incompatible with existing self-supervised models in the community. Second, the requirement for consistent task design across all candidate models restricts diversity in learned representations. These constraints not only limit the system's scalability but also hinder its ability to better diversify the feature representations. In future work, we are planning a more flexible routing framework that decouples task-specific components from the core architecture, enabling seamless integration of diverse self-supervised models while maintaining the performance guarantees.

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

## A  APPENDIX

### A.1  LLM USAGE

Large language models were used in this work to polish the writing.

### A.2  DETAILED BASE MODEL AND CLASSIFIER TRAINING

As we mentioned above, a variety of training strategies are implemented to encourage the development of diverse generalization abilities among the multiple base models. In addition to the original base model, we construct two other variant models. In the first variant, we employed random masking of the input. We generate a mask $m \in [0,1]^{C \times T}$ for each input $x \in R^{C \times T}$. And then, we obtain the mask sample $x_{mask} = x \odot m$, where $\odot$ represents element-wise multiplication. In the second variant, we added $L2$ regularization to the loss and dropout layer to the model structure. Each base model outputs three components: the latent representation, the reconstructed output, and the high-dimensional embedding, as follows:

$$\begin{aligned} h_1, p_1, z_1, \hat{z}_1 =& f_1(x), \\ h_2, p_2, z_2, \hat{z}_2 =& f_2(x_{mask}), \\ h_3, p_3, z_3, \hat{z}_3 =& f_3(x). \end{aligned} \quad (2)$$

The $f_1$, $f_2$ and $f_3$ denote the three base models, $h_1$, $h_2$ and $h_3$ denote the learned representations, $p_1$, $p_2$ and $p_3$ denote the reconstructed DE features, $z_1$, $z_2$ and $z_3$ denote the embeddings of reconstructions in the high-dimensional space, and $\hat{z}_1$, $\hat{z}_2$ and $\hat{z}_3$ denote the embeddings of ground truth in the high-dimensional space.

And then, the training objective of the base model is to minimize the combination of reconstruction and contrastive loss. Assume the training set has $N$ samples. Let $P_j = \{p_j^i | i = 1, 2, \ldots, N\}, j \in 1, 2, 3\}$ denote the reconstructions from the $j_{th}$ model, $Z_j = \{z_j^i | i = 1, 2, \ldots, N\}, j \in \{1, 2, 3\}$ denote the embeddings of reconstructions from the $j_{th}$ model, $\hat{Z}_j = \{\hat{z}_j^i | i = 1, 2, \ldots, N\}, j \in 1, 2, 3\}$ denote the embeddings of ground truth from the $j_{th}$ model, and $Y = \{y^i | i = 1, 2, \ldots, N\}$ denote the ground truth DE feature set.

$$L_1 = \min_{\theta_1} L_{mse}(P_1, Y) + \beta L_{cont}(Z_1, \hat{Z}_1),$$
$$L_2 = \min_{\theta_2} L_{mse}(P_2, Y) + \beta L_{cont}(Z_2, \hat{Z}_2), \tag{3}$$
$$L_3 = \min_{\theta_3} L_{mse}(P_3, Y) + \beta L_{cont}(Z_3, \hat{Z}_3) + \lambda \|\theta_3\|_2^2.$$

The $\theta_1$, $\theta_2$, and $\theta_3$ denote the parameters of the $f_1$, $f_2$, and $f_3$. $L_{mse}$ denotes the mean squared error (MSE), $L_{cont}$ denotes the contrastive loss, $\beta$ is a hyperparameter that controls the balance of reconstruction loss and contrastive loss, and $y$ denotes the ground truth DE feature of $x$.

As for the contrastive loss, for each model, we merge the two sets $Z_j$ and $\hat{Z}_j$ into a new set $\overline{Z}_j$ with a size of $2N$. In $\overline{Z}_j$, we combine reconstruction embedding $\overline{z}_j^{2k-1}$ and ground truth embedding $\overline{z}_j^{2k}$, $k \in \{1, \ldots, N\}$, as positive pairs. We use $K = \{1, \ldots, 2N\}$ to denote the index set. The $p_j(k)$ denotes the index of the positive sample of $\overline{z}_j^k$. Subsequently, the contrastive loss $L_{cont}$ is defined as shown in the following.

$$L_{cont}(Z_j, \hat{Z}_j) = -\sum_{k \in K} \log \frac{exp(\overline{z}_j^k \cdot \overline{z}_j^{p_j(k)}/\tau)}{\sum_{a \in A(k)} exp(\overline{z}_j^k \cdot \overline{z}_j^a/\tau)}, \tag{4}$$

where $A_j(k) = \{a | a \in K, \overline{z}_j^a \neq \overline{z}_j^k\}$, and $\tau$ is a scalar temperature parameter.

In stage 2, we set up a classifier for each base model to establish a mapping from the representations to the emotional labels.

$$e_j = \text{softmax}(clf_j(h_j)). \tag{5}$$

In this process, the base models are frozen, and the classifiers are trained to minimize the classification loss. Here, we adopt the cross-entropy loss to optimize the classifiers.

## A.3 Detailed Online Updating

We implement a 'model specialization and mutual assistance' style online parameter update mechanism. This mechanism can be regarded as a deepening of the multi-model online collaboration conception: (1) 'Specialization' identifies and promotes the most competent base models to refine their representational learning capabilities through test data streams; and (2) 'Mutual assistance' assists weaker classifiers in adapting to decision boundary drift by leveraging high-confidence predictors to propagate knowledge across the model ensemble. Collectively, this mechanism ensures synchronized evolution during deployment, substantially enhancing robustness to continuous distribution drifts.

We first introduce the base model parameter update strategy, i.e., model specialization. This approach encourages base models to further adapt to their respective specialized data distributions through online updates, thereby reinforcing the functional differentiation among them. For this reason, we adopt a multi-memory bank strategy that preserves distinct representational characteristics across models. Each base model $f_j$ maintains an independent memory bank $M_j^h$ implemented as a fixed-size FIFO queue with capacity $C_M$. This design ensures constant memory usage while recording the hard and most recent samples. When the $f_j$ is activated, the memory bank stores raw test EEG input $x'$ that represents hard samples into the $M_j^h$ during inference. We use the $M_s$ to denote the memory bank of the selected model. The memory update policy follows a selective strategy where only samples exceeding a dynamic threshold are included:

$$M_s^h \leftarrow \text{Push}\left(M_s^h, x'\right) \quad \text{if} \quad R_s > \gamma_s^{\text{dy}} \tag{6}$$

where $\gamma_s^{\text{dy}}$ represents the adaptive threshold for model $f_s$.

The threshold $\gamma_s^{\text{dynamic}}$ rules the selection of hard samples. It is computed dynamically based on each model's recent performance history as the activated base model. We maintain a sliding window $W_j$ that records the most recent $K$ routing scores for each model when it is activated. The adaptive threshold for the activated base model $f_s$ is computed as:

$$\gamma_s^{\text{dy}} = Q_{p1}(W_s), \tag{7}$$

where $Q_{p1}$ denotes the $p_1$-th percentile of the scores in the window. This adaptive threshold ensures that the system maintains sensitivity to distribution drifts while remaining robust to occasional outliers.

The fine-tuning process is initiated through an evidence-based triggering mechanism. Let $T_h$ denote the threshold number of new hard samples required to trigger fine-tuning. For each model $f_j$, we maintain a counter $c_j$ that increments when a new hard sample is added to $M_j^h$. Fine-tuning is triggered when sufficient evidence of distribution drift has been accumulated:

$$c_j \leftarrow c_j + 1, \quad \text{TriggerFineTune()} \quad \text{iff} \quad c_s \geq T_{h1}. \tag{8}$$

Upon triggering, $c_j$ is reset to zero, and the fine-tuning process randomly samples from $M_s^h$ to form the fine-tuning batch $B_{ft}^{enc}$. The contrastive loss is computed using the stored samples, same as Eq. equation 4:

$$
\begin{aligned}
Z', \hat{Z}' &\leftarrow f_s(B_{ft}^{enc}), \\
L_{ft}^{enc} &= L_{cont}(Z', \hat{Z}').
\end{aligned}
\tag{9}
$$

Our parameter unfreezing strategy adheres to the following principles. Since lower layers capture generic features, we primarily fine-tune higher-level components. To maintain computational efficiency and mitigate catastrophic forgetting, only a minimal set of parameters is updated. Furthermore, we selectively unfreeze the components most sensitive to current distributional drifts. Consequently, during fine-tuning, we unfreeze and update the following parameters:

$$\theta_s^{\text{enc}} = \{\theta_s^{\text{projector}}, \theta_s^{\text{norm}}, W_s^{out}, b_s^{out}\} \subset \theta_s. \tag{10}$$

These parameters are optimized using the Adam optimizer with a reduced learning rate $\alpha_{enc}^{\text{ft}}$:

$$\theta_s^{\text{enc}} \leftarrow \theta_s^{\text{enc}} - \alpha_{enc}^{\text{ft}} \cdot \nabla_{\theta_s^{\text{enc}}} L_{ft}^{enc} \tag{11}$$

Besides, routing scores are derived from the self-supervised task (reconstruction-contrastive) and are inherently tied to proxy tasks. They fail to directly reflect decision confidence in the downstream emotion classification task. Therefore, we also need to optimize classifiers online to address the issue of decision boundary adaptation. We introduce a mutual assistance mechanism that utilizes high-confidence predictions from strong models as pseudo labels to optimize other classifiers exhibiting prediction inconsistencies.

We retain both the logits and predicted labels from the selected classifier $clf_s$:

$$
\begin{aligned}
g_s &= clf_s(h_s), \\
e_s' &= \text{softmax}(g_s).
\end{aligned}
\tag{12}
$$

where $h_s$ is the representation from the $f_s$, $g_s$ is the logit vector, and $e_s'$ is the issued prediction. Following [NeurIPS20], we adopt an energy-based function to measure the confidence score of the selected classifier.

$$E_s = -T \cdot \log \sum_{c=1}^{C} \exp\left(\frac{g_s[c]}{T}\right), \tag{13}$$

where $C$ is the number of classes, $g_s[c]$ are the logits for class $c$ from the classifier $clf_s$, and $T$ is a temperature scaling factor. During training, we compute and store all the energy scores for three classifiers, $H_j^E$, in the training dataset. In the testing phase, the $W_s^E$ helps us identify the dynamic threshold for high-confidence prediction.

If the energy score $E_s$ falls below the $p_2$-th percentile of $H_s^E$, we consider the classifier's prediction to be highly confident.

$$E_s < \tau_s^{dy}, \tau_s^{dy} = Q_{p2}(H_s^E). \tag{14}$$

Then the predictions are used as pseudo-labels to supervise the online update of other classifiers. Analogous to the earlier design, we maintain a dedicated pseudo-label dynamic memory $M_j^p$ bank for each classifier. A consistency check is performed to identify the inconsistent classifiers. When a pseudo-label disagrees with a classifier's own prediction, the classifier is regarded as the inconsistent classifier. And the pseudo-label is pushed into its pseudo-label memory bank.

$$M_j^p \leftarrow \text{Push}(M_j^p, (e_s', h_j')) \quad \text{if} \quad e_j' \neq e_s', \tag{15}$$

where $e_j'$ are the predictions issued by the other $clf_j$, and $h_j'$ are the representations issued by the $f_j$.

Once a memory bank accumulates $T_{h2}$ new samples, we sample a batch $B_{ft}^{clf}$ from it and optimize the corresponding classifier $clf_j$ using the cross-entropy loss.

$$
\begin{aligned}
L_{clf}^{ft} &= \text{Crossentropy}(B_{ft}^{clf}(e_s'), clf_j(B_{ft}^{clf}(h_j'))), \\
\theta_j^{\text{clf}} &\leftarrow \theta_j^{\text{clf}} - \alpha_{clf}^{\text{ft}} \cdot \nabla_{\theta_j^{\text{clf}}} L_{ft}^{clf},
\end{aligned}
\tag{16}
$$

where $L_{ft}^{clf}$ is the loss for fine-tuning, and $\theta_j^{\text{clf}}$ are the parameters of the classifier $clf_j$.Merely activating appropriate models proves insufficient for long-term adaptation to data distribution drifts. We therefore require base models and classifiers to update their network parameters in response to the test data stream. To achieve this, we implement a 'model specialization and mutual assistance' style online parameter update mechanism. This mechanism can be regarded as a deepening of the multi-model online collaboration conception: (1) 'Specialization' identifies and promotes the most competent base models to refine their representational learning capabilities through test data streams; and (2) 'Mutual assistance' assists weaker classifiers in adapting to decision boundary drift by leveraging high-confidence predictors to propagate knowledge across the model ensemble. Collectively, this mechanism ensures synchronized evolution during deployment, substantially enhancing robustness to continuous distribution drifts.

We first introduce the base model parameter update strategy, i.e., model specialization. This approach encourages base models to further adapt to their respective specialized data distributions through online updates, thereby reinforcing the functional differentiation among them. For this reason, we adopt a multi-memory bank strategy that preserves distinct representational characteristics across models. Each base model $f_j$ maintains an independent memory bank $M_j^h$ implemented as a fixed-size FIFO queue with capacity $C_M$. This design ensures constant memory usage while recording the hard and most recent samples. When the $f_j$ is activated, the memory bank stores raw test EEG input $x'$ that represents hard samples into the $M_j^h$ during inference. We use the $M_s$ to denote the memory bank of the selected model. The memory update policy follows a selective strategy where only samples exceeding a dynamic threshold are included:

$$M_s^h \leftarrow \text{Push}\left(M_s^h, x'\right) \quad \text{if} \quad R_s > \gamma_s^{\text{dy}} \tag{17}$$

where $\gamma_s^{\text{dy}}$ represents the adaptive threshold for model $f_s$.

The threshold $\gamma_s^{\text{dynamic}}$ rules the selection of hard samples. It is computed dynamically based on each model's recent performance history as the activated base model. We maintain a sliding window $W_j$ that records the most recent $K$ routing scores for each model when it is activated. The adaptive threshold for the activated base model $f_s$ is computed as:

$$\gamma_s^{\text{dy}} = Q_{p1}(W_s), \tag{18}$$

where $Q_{p1}$ denotes the $p_1$-th percentile of the scores in the window. This adaptive threshold ensures that the system maintains sensitivity to distribution drifts while remaining robust to occasional outliers.

The fine-tuning process is initiated through an evidence-based triggering mechanism. Let $T_h$ denote the threshold number of new hard samples required to trigger fine-tuning. For each model $f_j$, we maintain a counter $c_j$ that increments when a new hard sample is added to $M_j^h$. Fine-tuning is triggered when sufficient evidence of distribution drift has been accumulated:

$$c_j \leftarrow c_j + 1, \quad \text{TriggerFineTune()} \quad \text{iff} \quad c_s \geq T_{h1}. \tag{19}$$

Upon triggering, $c_j$ is reset to zero, and the fine-tuning process randomly samples from $M_s^h$ to form the fine-tuning batch $B_{ft}^{enc}$. The contrastive loss is computed using the stored samples, same as Eq. equation 4:

$$
\begin{aligned}
Z', \hat{Z}' &\leftarrow f_s(B_{ft}^{enc}), \\
L_{ft}^{enc} &= L_{cont}(Z', \hat{Z}').
\end{aligned}
\tag{20}
$$

Our parameter unfreezing strategy adheres to the following principles. Since lower layers capture generic features, we primarily fine-tune higher-level components. To maintain computational efficiency and mitigate catastrophic forgetting, only a minimal set of parameters is updated. Furthermore, we selectively unfreeze the components most sensitive to current distributional drifts. Consequently, during fine-tuning, we unfreeze and update the following parameters:

$$
\theta_s^{\mathrm{enc}} = \{\theta_s^{\mathrm{projector}}, \theta_s^{\mathrm{norm}}, W_s^{out}, b_s^{out}\} \subset \theta_s.
\tag{21}
$$

These parameters are optimized using the Adam optimizer with a reduced learning rate $\alpha_{enc}^{\mathrm{ft}}$:

$$
\theta_s^{\mathrm{enc}} \leftarrow \theta_s^{\mathrm{enc}} - \alpha_{enc}^{\mathrm{ft}} \cdot \nabla_{\theta_s^{\mathrm{enc}}} L_{ft}^{enc}
\tag{22}
$$

Besides, routing scores are derived from the self-supervised task (reconstruction-contrastive) and are inherently tied to proxy tasks. They fail to directly reflect decision confidence in the downstream emotion classification task. Therefore, we also need to optimize classifiers online to address the issue of decision boundary adaptation. We introduce a mutual assistance mechanism that utilizes high-confidence predictions from strong models as pseudo labels to optimize other classifiers exhibiting prediction inconsistencies.

We retain both the logits and predicted labels from the selected classifier $clf_s$:

$$
\begin{aligned}
g_s &= clf_s(h_s), \\
e_s' &= \mathrm{softmax}(g_s).
\end{aligned}
\tag{23}
$$

where $h_s$ is the representation from the $f_s$, $g_s$ is the logit vector, and $e_s'$ is the issued prediction. Following [NeurIPS20], we adopt an energy-based function to measure the confidence score of the selected classifier.

$$
E_s = -T \cdot \log \sum_{c=1}^{C} \exp\left(\frac{g_s[c]}{T}\right),
\tag{24}
$$

where $C$ is the number of classes, $g_s[c]$ are the logits for class $c$ from the classifier $clf_s$, and $T$ is a temperature scaling factor. During training, we compute and store all the energy scores for three classifiers, $H_j^E$, in the training dataset. In the testing phase, the $W_s^E$ helps us identify the dynamic threshold for high-confidence prediction.

If the energy score $E_s$ falls below the $p_2$-th percentile of $H_s^E$, we consider the classifier's prediction to be highly confident.

$$
E_s < \tau_s^{dy}, \tau_s^{dy} = Q_{p_2}(H_s^E).
\tag{25}
$$

Then the predictions are used as pseudo-labels to supervise the online update of other classifiers. Analogous to the earlier design, we maintain a dedicated pseudo-label dynamic memory $M_j^p$ bank for each classifier. A consistency check is performed to identify the inconsistent classifiers. When a pseudo-label disagrees with a classifier's own prediction, the classifier is regarded as the inconsistent classifier. And the pseudo-label is pushed into its pseudo-label memory bank.

$$
M_j^p \leftarrow \mathrm{Push}(M_j^p, (e_s', h_j')) \quad \text{if} \quad e_j' \neq e_s',
\tag{26}
$$

where $e_j'$ are the predictions issued by the other $clf_j$, and $h_j'$ are the representations issued by the $f_j$.

Once a memory bank accumulates $T_{h2}$ new samples, we sample a batch $B_{ft}^{clf}$ from it and optimize the corresponding classifier $clf_j$ using the cross-entropy loss.

$$
\begin{aligned}
L_{clf}^{ft} &= \mathrm{Crossentropy}(B_{ft}^{clf}(e_s'), clf_j(B_{ft}^{clf}(h_j'))), \\
\theta_j^{\mathrm{clf}} &\leftarrow \theta_j^{\mathrm{clf}} - \alpha_{clf}^{\mathrm{ft}} \cdot \nabla_{\theta_j^{\mathrm{clf}}} L_{ft}^{clf},
\end{aligned}
\tag{27}
$$

where $L_{ft}^{clf}$ is the loss for fine-tuning, and $\theta_j^{\mathrm{clf}}$ are the parameters of the classifier $clf_j$.

## A.4 DETAILED BASE MODEL AND CLASSIFIER STRUCTURE

Our model consists of two main components: a Differential Entropy (DE) reconstruction module and a contrastive learning module. The DE reconstruction module aims to learn multi-band DE representations from raw EEG signals across multiple frequency bands. The contrastive learning module, on the other hand, is designed to align the learned representations with their corresponding ground truth DE values through an encoder-projector architecture based on self-attention.

The DE reconstruction module employs a multi-branch depth-wise 1-D convolutional structure with different kernel sizes to capture multi-scale features from the input raw EEG data $x$. Each branch processes the input using a depth-wise convolution.

$$g_b = \text{Conv1D}_{\text{depth-wise}}(x; k_b), \tag{28}$$

where $k_b$ denotes the kernel size of the $b$-th branch. After feature extraction, each branch is passed through a Multi-Layer Perceptron (MLP) to map the extracted features into DE values for that specific frequency band:

$$p_b = \text{MLP}(g_b), \tag{29}$$

where $p_b$ represents the reconstructed DE values for the $i$-th frequency band. Finally, the DE values from all branches are concatenated to form the final multi-band DE reconstruction:

$$p = concat(\{p_b\}), p \in R^{C \times d_b}, \tag{30}$$

where $d_b$ denotes the number of frequency bands.

The contrastive learning module takes both the reconstructed DE values $p$ and the ground truth DE values $y$ as inputs. It consists of an encoder and a projector. The encoder is built upon a self-attention mechanism. First, we perform the learnable position coding and linear layer mapping on the inputs:

$$E = p_{in} + P, \tag{31}$$

where $p_{in}$ is either reconstruction $p$ or ground truth $y$, and $P$ denotes the positional encoding matrix. Then, a self-attention mechanism is applied along the channel dimension to capture spatial dependencies:

$$
\begin{aligned}
Q &= EW_Q^\top, \quad K = EW_K^\top, \quad V = EW_V^\top; \\
A &= \text{Attention}(Q, K, V) = \text{softmax}\left(\frac{QK^\top}{\sqrt{d_k}}\right)V; \\
H_1 &= \text{LayerNorm}(A + E); \\
H_2 &= \text{ReLU}(H_1 W_{\text{ff1}} + b_{\text{ff1}})W_{\text{ff2}} + b_{\text{ff2}}; \\
H_3 &= \text{LayerNorm}(H_2 + H_1); \\
H &= H_3 W_{\text{out}} + b_{\text{out}}, W_{\text{out}} \in \mathbb{R}^{d_k \times d_b}.
\end{aligned} \tag{32}
$$

$W$ with various subscripts are learnable parameters, and $b$ with various subscripts are biases. $d_k$ is the hidden dimension.

Since the contrastive module takes both the reconstructed output and the ground truth as inputs, it generates two corresponding representations, $H$ and $\hat{H}$, respectively. An MLP architecture is utilized to project the learned representations $H$ and $\hat{H}$ into a high-dimensional space, resulting in embeddings $z$ and $\hat{z}$.

$$
\begin{aligned}
z_1 &= \text{ReLU}(H_{eith} W_1 + b_1), W_1 \in R^{(C \times d_b) \times d_{em1}}; \\
z_{eith} &= z_1 W_2 + b_2, W_2 \in R^{d_{em1} \times d_{em2}},
\end{aligned} \tag{33}
$$

where $H_{eith}$ is either $H$ or $\hat{H}$, and $z_{eith}$ is either $z$ or $\hat{z}$.

These two representations are then flattened and concatenated to form the final representation $h$.

We adopt a three-layer MLP as the classifier. The rectified linear unit (ReLU) activation functions are applied between each pair of consecutive layers. The numbers of hidden units in the two hidden layers are denoted as $d_{h1}$ and $d_{h2}$, respectively.

### A.5 IMPLEMENTATION DETAILS

To assess the inter-subject generalization performance of our method, we employ a leave-one-subject-out cross-validation (LOSOCV) evaluation protocol. In this protocol, data from one subject are used as the test set while the remaining data serve as the training set. This process is repeated iteratively so that each subject acts as the test set once. Final results are reported as the average accuracy and standard deviation across all folds, providing a robust estimate of subject-independent performance.

Table 3: The Performance of Each Base Model

| Method | Dreamer | | SEED |
| | Arousal | Valence | |
| --- | --- | --- | --- |
| Base model 1 | 66.10±6.35 | 67.01±7.39 | 82.18±6.97 |
| Base model 2 | 65.85±8.08 | 66.89±8.81 | 81.93±7.33 |
| Base model 3 | 66.38±6.98 | 67.79±7.77 | 83.39±6.87 |
| MMOC | 69.37±6.13 | 70.33±6.78 | 86.39±5.41 |

All experiments are conducted on an NVIDIA RTX 2080Ti GPU. During training, the temperature parameter $\tau$ is set to 0.07, L2 penalty $\lambda = 10^{-5}$, and the balancing factor $\beta = 0.9$. The DE features are calculated on the $\delta$, $\theta$, $\alpha$, $\beta$, and $\gamma$ frequency bands for SEED, and $\theta$, $\alpha$, and $\beta$ for Dreamer. Thus, the $d_b = 5$ for SEED, and the $d_b = 3$ for Dreamer. The kernel sizes of $k_b$ are $\in \{401, 51, 25, 15, 7\}$, corresponding to the five bands respectively. The $d_k = 64$, $d_{em1} = 256$, $d_{em2} = 128$, $d_{h1} = 62$, and $d_{h2} = 30$. The batch size is set to 128, and the base models are trained for 200 epochs with early stop. The base models are optimized using the Adam optimizer with a learning rate of 0.001. For the classification training, the Adam optimizer minimizes the cross-entropy loss with a learning rate of 0.0001 over 50 training epochs.

Additionally, we detail the hyperparameters governing the online inference and adaptation process. For the routing mechanism, the sliding window size for tracking recent scores $K$ is set to 32, and the percentile $p_1$ for calculating the dynamic threshold $\gamma_s^{dy}$ is 85. Each model's memory bank $M_s$ has a capacity $C_M$ of 16 for Dreamer and 32 for SEED. The fine-tuning process is triggered when the counter of new hard samples $c_s$ reaches the threshold $T_{h1} = 4$ for Dreamer and 8 for SEED, at which point a batch of $B_{ft} = 16$ for Dreamer and 32 for SEED samples is randomly drawn for encoder updating. The fine-tuning learning rate $\alpha_{enc}^{ft}$ is set to $5 \times 10^{-5}$. For classifier mutual assistance, the energy score percentile $p_2$ is 15, and the update threshold $T_{h2}$ is also 4 for Dreamer and 8 for SEED. The fine-tuning learning rate $\alpha_{clf}^{ft}$ is set to $1 \times 10^{-5}$.

### A.6 BASE MODEL PERFORMANCE EVALUATION

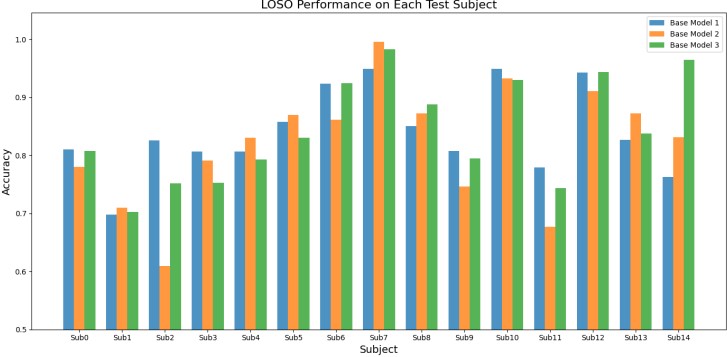

Figure 6: The LOSO performance of the base models on each test subjects.

In this work, we aim to propose a multi-model online collaboration framework for self-supervised learning along with its corresponding routing mechanism. To achieve this, we integrate multiple

base models into our framework. In this case, it is necessary to demonstrate that the performance improvements observed in MMOC stem from the online collaboration of multiple models rather than merely the robustness of individual base models. Consequently, we compare the performance of MMOC against each individual base model, as illustrated in Table 3.

As can be seen, the MMOC framework based on the three base models achieves better cross-subject performance than any individual model alone. This demonstrates that MMOC effectively integrates diverse base models and leverages its routing strategy to enhance performance on test samples.

Moreover, we present the accuracy of each base model across different test subjects on the SEED dataset in Figure 6. It can be observed that the base models exhibit obvious performance differences on some test subjects. For example, on Sub 2, Base Model 1 achieves the best performance, while Base Model 2 performs significantly worse than the others. On Sub 14, Base Model 3 yields the highest accuracy, Base Model 2 achieves the second-best result, and Base Model 1 performs the worst. This observation demonstrates that our diverse training strategy effectively differentiates the generalization capabilities of the base models. The observed variation in generalization performance across base models under different data distributions supports the rationale for model collaboration. By combining multiple models with complementary strengths, the framework achieves broader coverage of unknown distributions.

