# OpenReview forum: "MMOC: Multi-Model Online Collaboration Framework for Enhancing EEG Emotion Recognition"
_ICLR.cc/2026/Conference — Submitted to ICLR 2026_

### Official Review · Reviewer_7ZGR · 2025-10-23

**Soundness:** 2
**Presentation:** 2
**Contribution:** 2
**Rating:** 2
**Confidence:** 4

**Summary:**

This work introduces a self-supervised framework for EEG-based emotion recognition, termed **MMOC**. The proposed framework is characterized by three key features: **self-supervision**, **online learning**, and **multi-model collaboration**. The self-supervised design eliminates the need for annotated data, thereby reducing the dependence on costly manual labeling. The online learning capability allows the model to better adapt to real-world, continuously evolving environments. The collaborative mechanism among multiple models enhances generalization efficiency and enables the system to effectively cope with complex data drift. However, these advantages come with inherent challenges. For online learning, the primary challenge lies in developing an online update mechanism that allows network parameters to dynamically adapt to shifting data distributions. For multi-model collaboration, the main difficulty is defining a real-time, unsupervised metric to guide the routing process among models.

**Strengths:**

The topic of EEG-based emotion recognition is important and highly relevant to the advancement of brain–computer interfaces (BCIs).

**Weaknesses:**

1.The overall presentation of the work lacks logical coherence and is difficult to follow.

2.The methods section focuses primarily on describing how the model is constructed, without providing sufficient explanation of **why** such a design effectively achieves the stated objectives.

3.In Section 3, the text beginning with “Here is a fully paraphrased and academically refined version…” appears to be AI-generated and is inconsistent with the rest of the paper’s content.

The issues above are the main reasons for my decision to reject the paper.

**Questions:**

The training of each base model appears to be independent. How do the authors ensure that the resulting representations are aligned? While the paper claims that this design promotes diversity among base models, I believe it should aim to balance diversity **and** representational consistency to maintain effectiveness.

---

### Official Review · Reviewer_ebX4 · 2025-10-28

**Soundness:** 1
**Presentation:** 2
**Contribution:** 2
**Rating:** 2
**Confidence:** 5

**Summary:**

The paper introduces MMOC, a calibration-free framework for EEG emotion recognition under domain shift in online settings. It trains multiple base models with reconstruction and contrastive self-supervision, and at inference uses a routing mechanism that selects a model per sample based on normalized reconstruction and contrastive losses. Experiments on SEED and DREAMER under LOSO evaluation report higher accuracy than compared methods, and ablations indicate contributions from the routing and related components.

**Strengths:**

1\. The paper addresses an important and practical problem of calibration-free online EEG emotion recognition, which has been relatively less explored in prior research.

2\. The proposed method shows robustness under limited labeled data, demonstrating good performance even when annotation availability is reduced.

**Weaknesses:**

1. The paper’s organization could be improved. For instance, the title of Section 2.1 is placed incorrectly and should be moved further back by one paragraph. Some mathematical symbols (e.g., $L_j$, $L_{ce}$ in Figure 2) are used without explicit definition in the main text, which affects readability.

2. The framework involves multiple hyperparameters (e.g., τ, β, memory size, threshold quantiles), yet no sensitivity or robustness analysis is provided. This makes it difficult to assess the model’s stability.

3. While the method targets online EEG emotion recognition, the online update module still requires parameter updates that may be computationally costly. The paper needs to include additional experiments measuring inference and update latency to demonstrate that the system can operate in real-time conditions.

4. Only two datasets (SEED and DREAMER) are used for evaluation. Including at least one additional dataset would strengthen the empirical evidence and demonstrate broader applicability.

5. The experimental setup may not be entirely consistent with prior work. For example, SEED is treated as a binary task with 2-s segments, whereas some baselines (e.g., CLISA, GMSS, MSLTE) perform 3-class classification on 1-s segments. The paper should clarify which results are directly cited from prior publications and which are re-implemented under unified settings.

6. The framework employs three base models with relatively minor architectural differences (random masking, L2 + dropout variants). It remains unclear why structurally different models (e.g., CNN, RNN, Transformer) were not explored to provide stronger complementarity.

7. The pretraining stage combines reconstruction and contrastive learning, but the paper does not analyze the contribution of each pretraining component. It should also explain more clearly why the authors chose a multi-stage pretraining strategy instead of directly applying supervised learning with available labels.

8. In A.6 BASE MODEL PERFORMANCE EVALUATION, the paper reports per-subject results for each base model but does not report per-subject MMOC performance. The authors should add per-subject MMOC results to demonstrate that the proposed framework achieves consistent improvements across individual subjects.

**Questions:**

Please See Weakness.

---

### Official Review · Reviewer_GrEj · 2025-11-01

**Soundness:** 3
**Presentation:** 1
**Contribution:** 2
**Rating:** 4
**Confidence:** 4

**Summary:**

This paper addresses the challenge of inter-subject variability in online EEG data streams by introducing a self-supervised framework. Multiple base models are trained offline using reconstruction and contrastive losses on EEG data. During online inference, a routing mechanism selects the most appropriate model based on both reconstruction and contrastive losses. Online adaptation is further applied via memory-replay mechanisms. Experiments on SEED and Dreamer datasets demonstrate superior performance compared to several existing self-supervised approaches.

**Strengths:**

•	The paper tackles inter-subject variability in online EEG streams, which is a relevant problem for this conference.

•	The proposed framework empirically outperforms multiple state-of-the-art self-supervised EEG models.

**Weaknesses:**

While the paper addresses an important problem and demonstrates strong empirical performance, the paper presentation and several methodological and conceptual aspects need improvement.

The Experiments section begins with a drafting/AI-generation note (‘Here is a fully paraphrased and academically refined version of your provided text…’), which is inappropriate for a final manuscript. The construction and diversity of the model pool are not fully explained, raising questions about whether a single adaptively refined model could achieve comparable performance. The routing mechanism relies on reconstruction embeddings for contrastive loss, which may not reliably preserve semantic information and could lead to incorrect model selection. Scalability concerns arise because computing contrastive loss against all training embeddings for each test sample may be computationally expensive for real-time EEG inference. Key implementation details, including the choice of β to balance reconstruction and contrastive losses, selection of higher-level layers for fine-tuning, and sensitivity to these choices, are missing. The mutual-assistance mechanism could propagate errors if a high-confidence model is mistaken on out-of-distribution samples. Furthermore, the paper lacks qualitative evidence demonstrating that contrastive embeddings preserve emotion-relevant semantics during online updates and does not compare against recent online self-supervised learning methods as baselines such as:

o	Yu, X., Guo, Y., Gao, S., Simunic, T., 2022. SCALE: Online self-supervised lifelong learning without prior knowledge. CVPR 2023, 2484–2495.

o	Chen, T., Kornblith, S., Norouzi, M., Hinton, G.E., 2020. A simple framework for contrastive learning of visual representations. arXiv:2002.05709.

o	Duan, T., Wang, Z., Li, F., Doretto, G., Adjeroh, D. A., Yin, Y., & Tao, C. (2024). Online continual decoding of streaming EEG signal with a balanced and informative memory buffer. Neural Networks, 176, 106338

**Questions:**

1) How is the random input masking designed to preserve critical EEG semantics while still promoting model diversity?

2) Could a single adaptively refined model achieve comparable or superior performance compared to maintaining a pool of multiple models?

3) How were the base models in the pool constructed to have distinct generalization capabilities, and were they validated individually on test data streams?

4) The routing mechanism relies on reconstruction embeddings for contrastive loss. How reliable is this selection process in practice, especially if embeddings misalign with semantic information?

5)  Computing contrastive loss against all training embeddings for each test sample may be computationally expensive. How scalable is this approach for real-time EEG inference?

6) How is β chosen for balancing reconstruction and contrastive components?

7) What is the computational and storage cost of maintaining memory banks and computing energy-based pseudo-labels during online EEG streams?

8) How were the higher-level layers selected for fine-tuning, and how sensitive is performance to this choice?

9) Could the authors provide qualitative evidence, such as visualizations or embedding analyses, demonstrating that contrastive embeddings preserve emotion-relevant semantic information during online updates?

---

### Official Review · Reviewer_wujt · 2025-11-01

**Soundness:** 3
**Presentation:** 3
**Contribution:** 3
**Rating:** 6
**Confidence:** 4

**Summary:**

The paper introduces MMOC, anovel self-supervised framework
 with Multi-Model Online Collaboration (MMOC).For handling the varying input
 samples in the stream, MMOC proposes to activate the most suitable model from
 a candidate pool using arouting mechanism.This routing decision is guided by a
 hybrid reconstruction–contrastive performance, which comprehensively captures
 distribution drifts at both structural and semantic levels. Furthermore, each model
 is equipped with an online parameter update mechanism with models pecialization and mutual assistance. This mechanism not only enhances inter-model differentiation and specialization but also facilitates collaborative adaptation among
 models via pseudo-label sharing, thereby improving robustness against evolving
 data distributions. Extensive experiments on SEED and Dreamer datasets demonstrate that MMOC outperforms the state-of-the-art works.

**Strengths:**

1.This work proposes the MMOC framework, a self-supervised EEG emotion recognition
method based on multi-model online collaboration. By coordinating sample-level features
across multiple models for collaborative inference, the framework enhances generalization
capabilities and adaptability under distribution drift.

2.This work proposes a “model specialization and mutual assistance”-based online parameter
update mechanism. Leveraging the dynamic memory bank, this mechanism enhances
specialized representation learning while enabling cross-model mutual assistance through
the propagation of high-confidence pseudo labels.

**Weaknesses:**

Despite the excellent performance achieved in this paper, there is a lack of discussion on the time complexity comparison between multi-model and single-model approaches, which undermines the validity of its innovative points.

The experimental validation of the paper is insufficient.

1.The lack of experiments on time complexity makes it difficult to demonstrate real-time performance.

2. There is a lack of recent literature (published in the past two years) in the comparative experiments. It is recommended to add the following up-to-date comparative methods to enhance the comprehensiveness and timeliness of the experimental validation.

3.There is a lack of visual presentations for the ablation experiments. Additionally, the baseline model in the ablation experiments outperforms most comparative methods, and relevant discussions on this phenomenon are insufficient.

**Questions:**

Please refer to the issues mentioned in the weaknesses section.

---

### Meta-Review · Area_Chair_4z7A · 2026-01-03

**Summary:**

The reviewers consistently questioned the significance of the contribution and the sufficiency of the experimental evaluation

**Reviewer Scores:**

The reviewer scores remain unchanged

---

### Decision · Program_Chairs · 2026-01-26

Reject